# Online Algorithms for the Santa Claus Problem

**MohammadTaghi Hajiaghayi**
Department of Computer Science
University of Maryland
College Park, MD
hajiagha@umd.edu

**MohammadReza Khani**
Microsoft Bing Ads
Microsoft Corporation
Redmond, WA
khani87@gmail.com

**Debmalya Panigrahi**
Department of Computer Science
Duke University
Durham, NC
debmalya@cs.duke.edu

**Max Springer**
Department of Mathematics
University of Maryland
College Park, MD
mss423@umd.edu

## Abstract

The Santa Claus problem is a fundamental problem in *fair division*: the goal is to partition a set of *heterogeneous* items among *heterogeneous* agents so as to maximize the minimum value of items received by any agent. In this paper, we study the online version of this problem where the items are not known in advance and have to be assigned to agents as they arrive over time. If the arrival order of items is arbitrary, then no good assignment rule exists in the worst case. However, we show that, if the arrival order is random, then for $n$ agents and any $\varepsilon > 0$, we can obtain a competitive ratio of $1 - \varepsilon$ when the optimal assignment gives value at least $\Omega(\log n/\varepsilon^2)$ to every agent (assuming each item has at most unit value). We also show that this result is almost tight: namely, if the optimal solution has value at most $C \ln n/\varepsilon$ for some constant $C$, then there is no $(1 - \varepsilon)$-competitive algorithm even for random arrival order.

## 1   Introduction

Fair allocation of resources is one of the central themes of algorithmic fairness and game theory. In fact, the theory of fair division has its roots in mathematics going back to as early as 1948 [56]. In the general setting, this problem comprises a set of items that must be divided among a set of agents in an egalitarian manner, where each agent has a (possibly non-uniform) valuation for each item. A natural objective to capture the goal of fair division is to maximize the minimum total value of items received by any agent. This gives rise to the famous "Santa Claus problem" that we describe below.

In the Santa Claus problem, originally described by Bansal and Sviridenko in 2006 [12] (although it was studied under different names or assumptions prior to this), the Santa Claus is said to have a set of $m$ presents to be distributed equitably among $n$ children. Each child $i \in [n]$ has some arbitrary non-negative value $v_{ij}$ for present $j \in [m]$. Santa's goal is to distribute the presents in a way that makes the least satisfied child maximally satisfied. More formally, this means that the assignment seeks to maximize the minimum total value of the presents received by any child, where the total value of presents received by a child is the sum of her values for the presents that she received. The Santa Claus problem can be formalized as the following integer program:

$$\max \left\{ \min_{i \in [n]} \sum_{j=1}^{m} v_{ij} x_{ij} \;\middle|\; \sum_{i=1}^{n} x_{ij} \le 1 \;\forall j \in [m], \; x \in \{0,1\}^{mn} \right\}.$$

36th Conference on Neural Information Processing Systems (NeurIPS 2022).

There is substantial literature going back more than 50 years that studies variants of this problem in the offline setting (see related work). In many practical situations, however, the set of items to be allocated is not known in advance. For example, in online advertising, ad-space providers will receive monetary bids from competing agents for the display of their advertisements in real-time for an available space on a webpage [17, 15, 38, 59]. The provider must then make irrevocable decisions as to which advertiser's bid to accept based only on knowledge of prior allocations and the current bids for the available space [10, 22]. Beyond advertising, online allocation procedures have been useful in the study of donation distribution, wireless charging networks, organ donor matching, etc (see [4] for a survey of these applications).

Motivated by these applications, we consider the *online* Santa Claus problem in this paper. In this setting, the items arrive in an online sequence and must be allocated to one of the agents immediately upon arrival. As in the offline problem, our goal is to maximize the minimum total value among all agents. To illustrate the problem, consider the simple example in Figure 1 on the right where edges represent unit value of an agent for an item. At the time of the first arriving (leftmost) item, all agents can be matched to this item and therefore with probability $\frac{1}{3}$ any agent will receive it. However, as we continue forward with the input stream, we see in retrospect that the only nonzero max-min solution corresponds to the case where the first item was allocated to the rightmost agent (as it is the only item for which they have a nonzero value).

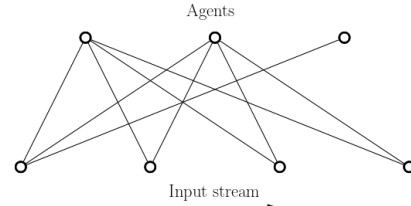

Figure 1: Example online problem instance. Fixed agents are presented in the top row and the arriving items are arranged from left to right in correspondence with their arrival order. An edge indicates a nonzero item valuation for a given agent.

Following standard terminology for online algorithms (see, e.g., [16]), we define the *competitive ratio* of an online algorithm as the minimum ratio between the value of the (maximization) objective in the algorithm's solution to that in the optimal (offline) solution in hindsight. We furthermore discuss the *additive regret* as the additive loss factor of our algorithm. More formally, we say our algorithm, ALG, has competitive ratio $c$ and additive regret $b$ if $\text{ALG} \geq c \cdot \text{OPT} - b$.

Prior work on the max-min objective in the online setting required various relaxations of the problem, such as allowing for some reordering in the allocation process [28], restricting the number of agents [39, 57, 58], or allowing migration of items after assignment [19]. This is because of two reasons. First, even in the offline setting, there remains a significant gap between the best upper and lower bounds on the approximation ratio of the Santa Claus problem, and bridging this gap is a major open problem. Second, as we will soon see, there is a simple construction for the online problem that shows the competitive ratio cannot be better than $n$. To bypass these bottlenecks, our first assumption in this paper is that the items arrive in *random order*. This is a standard assumption that has been used to simplify many related online problems [9, 24, 31, 32, 34, 41, 45]. But, even with this assumption, we show that obtaining a competitive solution is impossible in general *for small problem instances*. This motivates our second assumption: that the objective value of the optimal solution is sufficiently large (with respect to the values of individual items). With these two assumptions, we give an algorithm that obtains a competitive ratio of $(1 - \varepsilon)$ for any $\varepsilon > 0$. We note that using standard techniques, the assumption about the optimal objective being sufficiently large can be replaced by a corresponding additive regret in the competitive ratio.

We now formally define the two online input models that we consider in this paper: *adversarial* and *random order* input.

**Definition 1.1** (Adversarial Input). *An adversary selects the value vector $v \in [0,1]^n$ of each arriving item for all the agents, as well as the order in which these vectors arrive.*

**Definition 1.2** (Random-Order Input). *An adversary selects the value vector $v \in [0,1]^n$ of each arriving item for all the agents, but these vectors are randomly permuted to determine their arrival order.*

Note that in the literature, the independent and identically distributed (i.i.d.) input model is also often studied for related problems [34, 42, 49, 41, 2, 44, 55, 52]. In this model, the adversary picks a distribution over inputs that is unknown to the algorithm and arriving items are sampled i.i.d. from

Table 1: Online Results for the Santa Claus Problem

| Our Results | | |
|---|---|---|
| *Input Model* | *Algorithm* | *Competitive Ratio* |
| Adversarial | RANDOM | $\left(\frac{1-\varepsilon}{n}\right) \text{OPT} - O\left(\frac{n \log n}{\varepsilon^3}\right)$ |
| Random Order | GREEDYWR | $(1-\varepsilon)\text{OPT} - O\left(\frac{\log n}{\varepsilon}\right)$ |

this distribution. The random order model is *stronger* than the i.i.d. model in the sense that any algorithmic result for the random order model automatically extends to the i.i.d. setting as well. This includes the algorithmic results that we obtain in this paper for the random order arrival model.

## 1.1 Problem Definition

We here introduce the notation that we will use in the rest of the paper. Let $\mathbf{v}^1, ..., \mathbf{v}^m \in [0,1]^n$ denote the input sequence of items arriving in random order where $v_i^t$ is the value of the $t$-th item to the $i$-th agent. We additionally denote by $\mathbf{x}^1, ..., \mathbf{x}^m \in [0,1]^n$ the fractional allocation of each item by the algorithm. We further let the corresponding allocations of a fixed (offline) optimal solution be denoted as $\overline{\mathbf{x}}^1, ..., \overline{\mathbf{x}}^m \in [0,1]^n$ and let OPT denote the max-min objective value of the optimal solution. For simplicity, we slightly abuse notation by letting $\mathbf{V}^t = (v_1^t x_1^t, \ldots, v_n^t x_n^t)$ and $\overline{\mathbf{V}}^t = (v_1^t \overline{x}_1^t, \ldots, v_n^t \overline{x}_n^t)$ for $t \in [m]$. Our algorithm thus seeks to maximize the coordinate-wise minimum of $\sum_{t=1}^m \mathbf{V}^t$.

## 1.2 Our Contributions

First, we give a simple construction to show that if the arrival order of the items is adversarial, then the best competitive ratio that can be achieved is only $1/n$. (In fact, we also match this competitive ratio using a simple algorithm in the supplementary material.)

**Theorem 1.3** (Adversarial Input). *In the adversarial setting, no algorithm can obtain a competitive ratio better than* $1/n$.

This motivates us to consider the *random order model*, where an adversary again selects the set of items but they are then presented in random permutation order. For this setting, we give an algorithm that obtains a *fractional* assignment that is nearly optimal:

**Theorem 1.4** (Random Order: Algorithm). *For any $\varepsilon > 0$, there is an online fractional algorithm for the Santa Claus problem that has a competitive ratio of $(1 - \varepsilon)$ in the random order input model under the assumption that* $\text{OPT} \geq \Omega\left(\log n/\varepsilon^2\right)$.

We further show that, through randomized rounding, we can give an *integral* allocation that retains the near optimality of this fractional allocation.

Finally, we show that the lower bound on the value of OPT in the above theorem is *necessary*:

**Theorem 1.5** (Random Order: Impossibility Result). *For any $\varepsilon \in (0,1)$, there is no online algorithm for the Santa Claus problem in the random order input model that has a competitive ratio of $(1 - \varepsilon)$ when $\text{OPT} < C \cdot \frac{\ln n}{\varepsilon}$ for some (absolute) constant $C > 0$.*

The reader will note that the lower bound on OPT is precise as a function of $n$, but there is a slight mismatch between the upper and lower bounds as a function of $\varepsilon$ – bridging this gap is an interesting open question.

We summarize our results for the *online* Santa Claus problem in Table 1.2.

## 1.3 Related Work

The general case of the Santa Claus problem was initially explored (under a different name) in the field of algorithmic game theory for the fair allocation of goods [47]. By studying the assignment LP of [46] for the dual "makespan" problem, Bezakova and Dani [13] derived an additive approximation of $\max_{ij} v_{ij}$, i.e., the objective in the algorithm's solution is at least $\text{OPT} - \max_{ij} v_{ij}$ where

OPT is the objective value of the optimal solution. They also extended the hardness result on the dual makespan minimization problem [46] to demonstrate that the Santa Claus problem is NP-hard and cannot be approximated to a factor better than 2. Later, Bansal and Sviridenko [12] demonstrated that the integrality gap of the configuration LP for this problem is $\Omega(\sqrt{n})$, while Asadpour and Saberi [7] complimented this result with a $O(\sqrt{m}\log^3 m)$ upper bound for the same LP relaxation. To date, the best algorithmic result for the Santa Claus is an $\tilde{O}(n^\varepsilon)$-approximate algorithm, where $\varepsilon = \Omega(\log\log n/\log n)$, in quasi-polynomial time obtained by Chakrabarty, Chuzhoy, and Khanna [18].

For the special case of *restricted assignment*, i.e. $v_{ij} \in \{0, v_j\}$, Bansal and Sviridenko [12] provided an $\Omega(\log\log\log m/\log\log m)$-approximate algorithm that relies on rounding a configuration LP. Later, Feige gave a non-constructive proof that this LP relaxation was within a constant factor of OPT [29]. Asadpour, Feige, and Saberi [6] made this constructive, obtaining a $1/4$-approximation via a rounding algorithm based on local search, but the algorithm is not known to converge in polynomial time. Further work has since improved this constant factor [40, 20, 23, 5], improved upon the running time [21], and extended the setting beyond additive valuations [11].

**Online Assignment.** The study of online assignment is expansive, but much of the classical work is for adversarial arrival order. Even in the random order setting, a broad range of problems have been considered in recent years including the secretary problem [9, 45], AdWords [24, 31, 34], online matching [32, 41], online packing [37, 30, 44], online scheduling [51, 48], etc. One example of max-min online assignment in the random order setting is the work of Gollapudi and Panigrahi [36] who considered revenue maximization with fairness objectives. Another related work is that of Molinaro [51] for the dual min-max objective, who builds on prior work leveraging the experts framework from online learning [37] to give algorithms that simultaneously perform well in the adversarial and random-order settings. A third line of work relevant to our paper is that of online packing problems in the random arrival order (e.g., [30, 2, 44, 55, 52]. In particular, the results of Agrawal *et al.* [2] have a similar flavor to ours: they obtain $(1 - \varepsilon)$-competitiveness assuming a large enough optimal value for the online packing problem in the random order setting.

## 2 Online Algorithm for the Santa Claus Problem in the Random Order Model

In this section, we present the approximately greedy algorithm, Algorithm 1, and analyze its competitive ratio in the random order model. Building on the work of Molinaro for online scheduling [51], we use a greedy algorithm for a smoothed version of our objective function and a restart procedure during the online allocation process to reduce the impact of correlations that arise in this input model.

A natural strategy for our problem is to allocate the arriving item to the least satisfied agent. However, one can show this strategy has too high an additive regret [37, 51] as the change in our solution value can vary quickly from one iteration to the next. Instead, we use a *smoothed* version of the greedy algorithm. The algorithm is designed as follows: we first define $\phi_\varepsilon$ to be a re-scaled variant of the LOGSUMEXP function that serves as a smoothed minimum. For the first half of the input stream, we select an allocation for each arriving item that maximizes the increase in our smoothed objective function. This stage can be thought of as approximately greedy with respect to the *gradient* of $\phi_\varepsilon$. After $\frac{m}{2}$ items have been allocated, we "restart" the allocation by maximizing the increase in our objective with respect to the $t > \frac{m}{2}$ allocations only. This restart procedure is essential for reducing the correlations that arise in sampling without replacement in the random order model since at each iteration of the allocation procedure, our decision depends on at most $\frac{m}{2} - 1$ items. The pseudocode of this procedure is presented in Algorithm 1.

### 2.1 Algorithm Analysis

In the analysis of the competitive ratio of Algorithm 1, we will leverage several key facts about the smoothed minimum function $\phi_\varepsilon$. This smoothness implies that the gradient nicely captures incremental increase in the objective function, thus allocating with respect to this produces an essentially greedy process, allowing us to follow the analysis of [1, 24, 51] to get the desired guarantees for the random-order model. We now state our main result in Theorem 2.1.

---

**ALGORITHM 1:** SMOOTH GREEDY WITH RESTART

---

**Input:** $0 < \varepsilon < 1$, input stream $\mathcal{M}$ of $m$ items

Define $\phi_\varepsilon(u) = -\frac{1}{\varepsilon} \ln \left( \sum_i e^{-\varepsilon u_i} \right)$;

**for** $t = 1$ *to* $m/2$ **do**

    Select $\mathbf{x}^t \in \Delta^n$ to maximize $\phi_\varepsilon \left( \sum_{\tau=1}^{t} \mathbf{V}^\tau \right)$;

**end**

**for** $t = m/2 + 1$ *to* $m$ **do**

    Select $\mathbf{x}^t \in \Delta^n$ to maximize $\phi_\varepsilon \left( \sum_{\tau=\frac{m}{2}+1}^{t} \mathbf{V}^\tau \right)$;

**end**

---

**Theorem 2.1.** *For any $\varepsilon > 0$, Algorithm 1 guarantees in the random-order input model that the expected value of the allocation assigned to any agent is at least*

$$(1 - \varepsilon) \cdot \mathrm{OPT} - O\left( \frac{\log n}{\varepsilon} \right).$$

Note that this theorem immediately implies the following corollary since the additive regret term can be absorbed in the multiplicative error for sufficiently large OPT:

**Corollary 2.2.** *For any $\varepsilon > 0$, Algorithm 1 has a competitive ratio of $(1 - \varepsilon)$ for $\mathrm{OPT} \geq \Omega(\log n / \varepsilon^2)$.*

In order to prove this theorem, we first show some properties of the smoothed minimum function $\phi_\varepsilon$ utilized by Algorithm 1. We will then prove some technical lemmas that will help establish the theorem.

Lemma 2.3 effectively defines the additive error with respect to our true objective, the agent-wise minimum, and stability of the smooth function following each allocation decision. As was shown in prior work, allocating with respect to the order statistics or even the $L_p^p$ norm produces either too high of a regret factor, or instability in the derivative value under small perturbations to the input value. As such, the smoothing and the following properties are critical to maintaining our regret and competitive ratio bounds.

**Lemma 2.3.** *For all $u \in \mathbb{R}^n$, $v \in [0,1]^n$, and $\varepsilon > 0$, the function $\phi_\varepsilon(x) = -\frac{1}{\varepsilon} \ln \left( \sum_{i=1}^{n} e^{-\varepsilon x_i} \right)$ satisfies the following:*

*(a)* $\min_i \{u_i\} - \frac{\ln n}{\varepsilon} \leq \phi_\varepsilon(u) < \min_i \{u_i\}$

*(b)* $\nabla \phi_\varepsilon(u + v) \in e^{\pm \varepsilon} \cdot \nabla \phi_\varepsilon(u)$

*Furthermore, if $u_i \geq v_i$ for each $i \in [n]$, we have*

*(c)* $\phi_\varepsilon(u - v) \leq \phi_\varepsilon(u) - \phi_\varepsilon(v)$

The proof of these properties are deferred to Appendix A due to space constraints.

Now utilizing these two properties we can prove the following important bound on the inner product of the smoothed minimum's gradient. This will be used throughout our analysis to bound the incremental change in the objective MAXMIN value after each allocation decision, and the summation of these changes can be seen as the accumulated "reward" at any given stage of the input stream. The proof is by direct integration of stability property $(b)$ and is thus deffered to Appendix A.

**Lemma 2.4.** *For $u \in \mathbb{R}^n$ and $v, v' \in [0,1]^n$, if $\phi_\varepsilon(u + v) \geq \phi_\varepsilon(u + v')$ then $\langle \nabla \phi_\varepsilon(u), v \rangle \geq e^{-2\varepsilon} \langle \nabla \phi_\varepsilon(u), v' \rangle$.*

We now follow in the intuition of Agrawal and Devanur [1] to prove Theorem 2.1 by bounding the incremental increase in our "reward" both before and after the restart at $t = \frac{m}{2}$. By implementing this restart in the allocation procedure, we segment the stream into two portions that have identical probabilistic guarantees and reduce the correlation between input elements to allow for an optimal competitive ratio and low additive regret.

*Proof of Theorem 2.1.* We note again that in the approximately greedy procedure of Algorithm 1, we are essentially seleting items greedily according to the *gradient* of $\phi_\varepsilon$ to maximize the incremental changes. Due to the restart at $m/2$, we define

$$\nabla^t = \nabla\phi_\varepsilon\left(\sum_{\tau=1}^{t-1}\mathbf{V}^\tau\right) \text{ for } t \leq \frac{m}{2} \text{ and } \nabla^t = \nabla\phi_\varepsilon\left(\sum_{\tau=m/2+1}^{t-1}\mathbf{V}^\tau\right) \text{ for } t > \frac{m}{2}$$

We now proceed by deriving a bound on $\min_i\{\sum_{\tau=1}^m\mathbf{V}_i^\tau\}$ in terms of our smoothed-approximation function $\phi_\varepsilon$, thus bounding the accumulated error by the algorithm when estimating our true objective. By the concavity of $\phi_\varepsilon$ and Lemma 2.4, we have that

$$\phi_\varepsilon\left(\sum_{\tau=1}^t\mathbf{V}^\tau\right) - \phi_\varepsilon\left(\sum_{\tau=1}^{t-1}\mathbf{V}^\tau\right) \geq \left\langle\nabla\phi_\varepsilon\left(\sum_{\tau=1}^t\mathbf{V}^\tau\right),\mathbf{V}^t\right\rangle \qquad \text{(Concavity)}$$

$$\geq e^{-\varepsilon}\left\langle\nabla\phi_\varepsilon\left(\sum_{\tau=1}^{t-1}\mathbf{V}^\tau\right),\mathbf{V}^t\right\rangle = e^{-\varepsilon}\langle\nabla^t,\mathbf{V}^t\rangle. \quad \text{(Lemma 2.4)}$$

Without loss of generality we proceed by considering the first half of the input sequence ($t \leq \frac{m}{2}$). By summing this inequality over the input prior to the restart (from $t = 1$ to $\frac{m}{2}$), we have

$$\phi_\varepsilon\left(\sum_{\tau=1}^{m/2}\mathbf{V}^\tau\right) - \phi_\varepsilon(0) = \phi_\varepsilon\left(\sum_{\tau=1}^{m/2}\mathbf{V}^\tau\right) + \frac{\ln n}{\varepsilon} \geq e^{-\varepsilon}\sum_{t=1}^{m/2}\langle\nabla^t,\mathbf{V}^t\rangle \qquad (1)$$

Now, taking into account the allocation before and after the restart at $\frac{m}{2}$ and invoking Eq. 1 for each half of the input stream with the concavity of $\phi_\varepsilon$, we obtain

$$\sum_{t=1}^m\langle\nabla^t,\mathbf{V}^t\rangle \leq e^\varepsilon\left(\phi_\varepsilon\left(\sum_{\tau=1}^{m/2}\mathbf{V}^\tau\right) + \phi_\varepsilon\left(\sum_{\tau=m/2+1}^m\mathbf{V}^\tau\right) + \frac{2\ln n}{\varepsilon}\right) \qquad \text{(Ineq. 1)}$$

$$\leq e^\varepsilon\left(\phi_\varepsilon\left(\sum_{\tau=1}^m\mathbf{V}^\tau\right) + \frac{2\ln n}{\varepsilon}\right) \qquad \text{(Lemma 2.3c)}$$

$$\leq e^\varepsilon\left(\min_i\{\sum_{\tau=1}^m\mathbf{V}_i^\tau\} + \frac{2\ln n}{\varepsilon}\right) \qquad \text{(Lemma 2.3a)}$$

Lastly, by rearranging terms we can bound the element-wise minimum as

$$\min_i\{\sum_{\tau=1}^m\mathbf{V}_i^\tau\} \geq e^{-\varepsilon}\left(\sum_{t=1}^m\langle\nabla^t,\mathbf{V}^t\rangle\right) - \frac{2\ln n}{\varepsilon}. \qquad (2)$$

Thus, the output of our algorithm will approximate the *actual* objective function within a multiplicative factor of $e^{-\varepsilon} \approx 1 - \varepsilon$ and an additive regret on the order of $\ln n/\varepsilon$.

We now proceed to bound the gap between our algorithmic solution to that of the optimal *offline* solution which, combined with the above error, will give the final result. As such, the final step of our analysis will be to take the expectation of this inequality to get the final bounds in Theorem 2.1. Due to the restart and random order input model, the expected increase in maximal value over the first and second half is equivalent,

$$\mathbb{E}\left[\sum_{t=1}^{m/2}\langle\nabla^t,\mathbf{V}^t\rangle\right] = \mathbb{E}\left[\sum_{t=m/2+1}^m\langle\nabla^t,\mathbf{V}^t\rangle\right]$$

so without loss of generality we need only bound the first half's value and apply this bound to both portions. The benefit of this restart will yield a tolerable error as compared to the optimal offline solution in each half of the allocation procedure, rather than a continuously accumulating divergence between the two solutions: each arriving job's allocation is only dependent upon (at most) $\frac{n}{2} - 1$ other decisions. We leverage this randomness to obtain the final competitive guarantees [52].

Now, by the nature of greedy selection to maximize our objective function, we must have that in each iteration our algorithm's selection produces an increase in value that is at least as good as that of the optimal offline solution: $\phi_\epsilon(\sum_{\tau=1}^{t} \mathbf{V}^\tau) \geq \phi_\epsilon(\sum_{\tau=1}^{t-1} \mathbf{V}^\tau + \overline{\mathbf{V}}^t)$ for each $t$[1]. Combining this with Lemma 2.4, we must have

$$\langle \nabla^t, \mathbf{V}^t \rangle \geq e^{-2\varepsilon} \langle \nabla^t, \overline{\mathbf{V}}^t \rangle$$

Furthermore, since $\nabla\phi_\varepsilon \in \ell_1^+$ [33] and $\nabla^t$ is independent of $\mathbf{V}^t$, we prove the following purely probabilistic result that in Appendix A will be used to bound the above summations to derive the final result.

**Lemma 2.5.** *Consider a set of vectors $\{y^1, ..., y^m\} \in [0,1]^n$ and let $\{\mathbf{Y}^i\}_{i=1}^{k}$ be sampled without replacement. Let $\mathbf{Z} \in \ell_1^+$ be a random vector that depends only on $\{\mathbf{Y}^i\}_{i=1}^{k-1}$. Then for all $\varepsilon > 0$,*

$$\mathbb{E}\langle \mathbf{Y}^k, \mathbf{Z} \rangle \geq e^{-\varepsilon} \min_i \{\mathbb{E}\mathbf{Y}_i^k\} - \frac{\ln n}{\varepsilon (m - k + 1)}.$$

Using this lower bounding result, we can now bound the sum of rewards as

$$\mathbb{E}\left[\langle \nabla^t, \overline{\mathbf{V}}^t \rangle\right] \geq e^{-\varepsilon} \min_i \{\mathbb{E}\left[\overline{\mathbf{V}}_i^t\right]\} - \frac{\ln n}{\varepsilon(m - t + 1)}. \tag{3}$$

Adding this inequality over all $t \leq \frac{m}{2}$ in combination with $\min_i\{\mathbb{E}[\overline{\mathbf{V}}^t]_i\} = \frac{\text{OPT}}{m}$ we conclude that

$$e^{2\varepsilon} \cdot \mathbb{E}\sum_{t \leq \frac{m}{2}} \langle \nabla^t, \mathbf{V}^t \rangle \geq \mathbb{E}\sum_{t \leq \frac{m}{2}} \langle \nabla^t, \overline{\mathbf{V}}^t \rangle \qquad \text{(Lemma 2.3)}$$

$$\geq e^{-\varepsilon} \sum_{t \leq \frac{m}{2}} \min_i \{\mathbb{E}\left[\overline{\mathbf{V}}_i^t\right]\} - \sum_{t \leq \frac{m}{2}} \frac{\ln n}{\varepsilon(m - t + 1)} \qquad \text{(Ineq. 3)}$$

$$= e^{-\varepsilon}\left(\frac{\text{OPT}}{2}\right) - \frac{\ln n}{\varepsilon}$$

Due to the restart at $t = \frac{m}{2}$, we can extend the sum to $t = m$ by simply doubling the above RHS. Finally, invoking inequality (2), we see that

$$\mathbb{E}\left[\min_i \sum_{\tau=1}^{m} \mathbf{V}^\tau\right] \geq e^{-4\varepsilon}\text{OPT} - O\left(\frac{\log n}{\varepsilon}\right) \geq (1 - O(\varepsilon))\,\text{OPT} - O\left(\frac{\log n}{\varepsilon}\right).$$

by the Taylor approximation $e^{-x} \geq 1 - x$. $\qquad\qquad\square$

## 2.2 Online Rounding Algorithm

The previous algorithm produces an online fractional solution for the Santa Claus problem. We now show that using simple randomized rounding, we can convert this into an integer solution.

**Theorem 2.6.** *Fix any $\varepsilon > 0$. Given a $(1 - \varepsilon)$competitive online fractional algorithm for the Santa Claus problem, there is an online (integral) algorithm whose competitive ratio is $(1 - O(\varepsilon))$, provided* $\text{OPT} \geq \Omega\left(\frac{\log n}{\varepsilon^2}\right)$.

*Proof.* The algorithm is simply randomized rounding. If an item $j$ is allocated with fraction $x_{ij}$ to agent $i$ such that $\sum_{i=1}^{n} x_{ij} \leq 1$ by the fractional solution, then we assign item $i$ to agent $j$ with probability $x_{ij}$. Note that since $v_{ij} \in [0,1]$, the fractional value derived by an agent $i$ from an item $j$, given by $v_{ij}x_{ij}$ is also in $[0,1]$. Thus, by Chernoff bounds, the probability that the total value of agent $i$ in the rounded assignment is less than $(1 - \varepsilon)\sum_{j=1}^{m} v_{ij}x_{ij}$ is at most

$$\exp\left(-\frac{\varepsilon^2}{3} \cdot \sum_{j=1}^{m} v_{ij}x_{ij}\right) \leq \exp\left(-\frac{\varepsilon^2}{3} \cdot (1 - \varepsilon) \cdot \text{OPT}\right) \leq \frac{1}{n^2}, \text{ for OPT} \geq \Omega\left(\frac{\log n}{\varepsilon^2}\right).$$

---

[1]The optimal offline algorithm may allocate in a manner that does not maximize the objective function's increase at iteration $t$ in anticipation of better allocation options later in the input sequence.

Thus, with probability at least $1 - \frac{1}{n^2}$, we have

$$(1 - \varepsilon) \sum_{j=1}^{m} v_{ij} x_{ij} \geq (1 - \varepsilon)^2 \cdot \text{OPT} > (1 - 2\varepsilon) \cdot \text{OPT}.$$

It follows that the probability that the total value of agent $i$ in the rounded assignment is less than $(1 - 2\varepsilon) \cdot \text{OPT}$ is at most $\frac{1}{n^2}$. Using the union bound over the $n$ agents, we get that the probability that the total value of *any* agent in the rounded assignment is less than $(1 - 2\varepsilon) \cdot \text{OPT}$ is at most $\frac{1}{n}$. Thus, the expected value of the objective is at least $(1 - \frac{1}{n})(1 - 2\varepsilon) \cdot \text{OPT} > (1 - 3\varepsilon) \cdot \text{OPT}$ for large enough $n$. $\qquad\square$

# 3 Random Order Lower Bound

We here present an impossibility result on the Santa Claus problem under the random order input models.

**Theorem 3.1.** *For any $\gamma \in (0, 1)$, if a randomized algorithm* ALG *for the online Santa Claus problem with random order input satisfies*

$$\text{ALG} \geq (1 - \gamma) \cdot \text{OPT},$$

*then* $\text{OPT} \geq \frac{C \ln n}{\gamma}$ *for some (absolute) constant $C > 0$.*

We use the following construction from the proof of Theorem 1.3. There are $n$ agents, of which $n - 1$ are *private* agents and the remaining one is a *public* agent. Every private agent has $k$ distinct private items for which their valuation is 1 each, and the valuation for every other agent 0. In addition, there are $k$ public items, each of which has a valuation of 1 for every agent. The optimal solution is to assign the private items to the corresponding private agents, and the public items to the public agent. Thus, $\text{OPT} = k$.

Now, when the items are presented in uniform random order, consider the first $\varepsilon$ fraction of presented items – call this the $\varepsilon$-prefix. Our main claim is that with constant probability, the following statements both hold:

   (a)  there are about $\varepsilon$ fraction of public items in this $\varepsilon$-prefix

   (b)  there is at least one type of private item that is missing from this $\varepsilon$-prefix.

For property (a), we need the following concentration inequality from Devanur and Hayes [24]:

**Lemma 3.2** (Lemma 3 in [24]). *Let $Y = (Y_1, \ldots, Y_m)$ be a vector of real numbers, and let $\varepsilon \in (0, 1)$. Let $S$ be a random subset of $[m]$ of size $\varepsilon m$, and set $Y_S := \sum_{j \in S} Y_j$. Then, for every $\delta \in (0, 1)$,*

$$\mathbf{Pr}\left[ |Y_S - \mathbb{E}[Y_S]| \geq \frac{2}{3} \|Y\|_\infty \ln\left(\frac{2}{\delta}\right) + \|Y\|_2 \sqrt{2\varepsilon \ln\left(\frac{2}{\delta}\right)} \right] \leq \delta.$$

Property (a) concerning the fraction of public items in the $\varepsilon$-prefix follows almost immediately from the above lemma:

**Lemma 3.3.** *The probability that there are fewer than $\frac{5\varepsilon}{12}$ fraction of public items in the $\varepsilon$-prefix is at most $2\exp\left(-\varepsilon k / 8\right)$.*

*Proof.* We invoke 3.2 with the following setting of variables. Let $Y$ be a binary vector where $Y_i = 1$ for $i \in [k]$ and $Y_i = 0$ otherwise. ($Y_i$ is the indicator for whether an item is a public item.) $S$ represents the set of indices in the $\varepsilon$-prefix. Then, $Y_S$ counts the number of public items in the $\varepsilon$-prefix in a random ordering of the items. Clearly, $\mathbb{E}[Y_S] = \varepsilon k$, $\|Y\|_\infty = 1$, and $\|Y\|_2 = \sqrt{k}$. Finally, set $\delta = 2e^{-\frac{\varepsilon k}{8}}$, or more so $\ln\left(\frac{2}{\delta}\right) = \frac{\varepsilon k}{8}$. Now, by 3.2, we have

$$\mathbf{Pr}\left[ |Y_S - \varepsilon k| \geq \frac{2}{3} \cdot \frac{\varepsilon k}{8} + \sqrt{k} \cdot \sqrt{2\varepsilon \cdot \frac{\varepsilon k}{8}} \right] = \mathbf{Pr}\left[ |Y_S - \varepsilon k| \geq \frac{7}{12} \cdot \varepsilon k \right] \leq 2e^{-\frac{\varepsilon k}{8}}.$$

and the lemma follows. $\qquad\square$

Now, we show property (b) to demonstrate the non-existence of at least one type of private item in the $\varepsilon$-prefix.

**Lemma 3.4.** *The probability that all the $n-1$ types of private items appear in the $\varepsilon$-prefix is at most* $\exp\left(-(n-1) \cdot 4^{\frac{-\varepsilon k}{1-\varepsilon}}\right)$.

*Proof.* Fix a type of private item, say those of type-$j$, i.e. only agent $j$ (for some $j \in [n-1]$) has unit value for this type of item while all other agents have $0$ value. First, we bound the probability that no item of type-$j$ appears in the $\varepsilon$-prefix. To do this, note that this probability can be written, using the chain rule for conditional probabilities, as the product (over $i$ from $1$ to $\varepsilon n k$) of the probabilities of the $i$th item in the $\varepsilon$-prefix not being a type-$j$ item under the condition that the first $i-1$ items were not type-$j$ items either. Clearly, this probability, for any $i \leq \varepsilon n k$, is at least $1 - \frac{1}{(1-\varepsilon)n}$ since there are $k$ items of type-$j$ among at most $(1-\varepsilon)nk$ items overall after the conditioning. Thus, the probability that no item of type-$j$ appears in the $\varepsilon$-prefix is at least

$$\left(1 - \frac{1}{(1-\varepsilon)n}\right)^{\varepsilon n k} \geq 4^{\frac{-\varepsilon}{1-\varepsilon} \cdot k} \text{ by choosing } n \geq \frac{2}{1-\varepsilon}.$$

Denote $p = 4^{\frac{-\varepsilon}{1-\varepsilon} \cdot k}$. Consider the events that at least one item of type-$j$ appears in the $\varepsilon$-prefix. These events are negatively correlated and therefore, the probability that at least one item of type-$j$ appears in the $\varepsilon$-prefix for every $j \in [n-1]$ is at most

$$(1-p)^{n-1} \leq \left(1 - 4^{\frac{-\varepsilon}{1-\varepsilon} \cdot k}\right)^{n-1} \leq e^{-4^{\frac{-\varepsilon}{1-\varepsilon} k}(n-1)}.$$

$\qquad\square$

Now, by setting $k = \frac{1-\varepsilon}{2\varepsilon} \cdot \lg(n-1)$ we obtain

$$(n-1) \cdot 4^{\frac{-\varepsilon}{1-\varepsilon} k} = (n-1) \cdot 2^{-\frac{2\varepsilon}{1-\varepsilon} \cdot \frac{1-\varepsilon}{2\varepsilon} \cdot \lg(n-1)} = (n-1) \cdot 2^{-\lg(n-1)} = 1,$$

and furthermore,

$$\frac{\varepsilon k}{8} = \frac{1-\varepsilon}{16} \cdot \lg(n-1).$$

Plugging these expressions into 3.3 and 3.4, we get that the probability of every private item type appearing in the $\varepsilon$-prefix is at most $\frac{1}{e}$ and the probability of fewer than $\frac{5\varepsilon}{12}$ fraction of public items appearing in the $\varepsilon$-prefix is at most $2e^{-\frac{1-\varepsilon}{16} \cdot \lg(n-1)}$. The latter probability can be further simplified to

$$2e^{-\frac{1-\varepsilon}{16} \cdot \lg(n-1)} = 2e^{-\frac{1-\varepsilon}{16} \cdot \frac{\ln(n-1)}{\ln 2}} = \left(\frac{2}{n-1}\right)^{\frac{1-\varepsilon}{16 \ln 2}} < \frac{1}{e} \text{ for large enough } n.$$

Using the union bound over these two events, we conclude that with probability at least $1 - \frac{2}{e}$, there is at least one missing private item type in the $\varepsilon$-prefix, and also at least $\frac{5\varepsilon}{12}$ fraction of private items appear in the $\varepsilon$-prefix. In this case, in the $\varepsilon$-prefix, the algorithm cannot distinguish between the private agent whose item type is missing and the public agent. Thus, in expectation (over the randomness of the algorithm), at least half the public items in the $\varepsilon$-prefix are assigned to the private agent whose item type is missing. As a consequence, at least $\frac{5\varepsilon}{24}$ fraction of the public items are not allocated to the public agent in the entire algorithm, which means that the total valuation of the public agent is at most $\left(1 - \frac{5\varepsilon}{24}\right)k$. Thus, in order to guarantee

$$\text{ALG} \geq (1-\gamma) \cdot \text{OPT},$$

we need $\gamma \geq \frac{5\varepsilon}{24}$, i.e., $\frac{1}{\varepsilon} \geq \frac{5}{24\gamma}$. This implies by the set value of $k$ that

$$\text{OPT} = k = \frac{1-\varepsilon}{2\varepsilon} \cdot \lg(n-1) = \frac{\frac{1}{\varepsilon} - 1}{2} \cdot \lg(n-1) \geq \frac{\frac{5}{24\gamma} - 1}{2} \cdot \lg(n-1) \geq C \cdot \frac{\ln n}{\gamma} \text{ for some constant } C.$$

This completes the proof of 3.1. $\qquad\square$

In the current work, we have presented impossibility results for the online Santa Claus problem in adversarial and random order input models, as well as a near optimal algorithm for the random order setting. These results effectively address the necessary assumptions on the problem to obtain optimal online solutions, and furthermore obtain these results via simplistic algorithms. Since the MAXMIN objective function is the dual of MINMAX used in the makespan and load balancing literature, we hope that the impossibility results presented here can be carried over to address the remaining open questions regarding their tightest additive terms in competitive ratio analysis.

# 4 Acknowledgements and Disclosure of Funding

MohammadTaghi Hajiaghayi was upported in part by NSF CCF grants 2114269 and 2218678. Debmalya Panigrahi was supported in part by NSF grants CCF-1750140 (CAREER) and CCF-1955703 and ARO grant W911NF2110230. Max Springer was supported by the National Science Foundation Graduate Research Fellowship Program under Grant No. DGE 1840340. Any opinions, findings, and conclusions or recommendations expressed in this material are those of the author(s) and do not necessarily reflect the views of the National Science Foundation.

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
