# Appendix

## A  Omitted Proofs

### A.1  Online Santa Claus with Adversarial Arrival Order

*Proof of Theorem 1.3.* We consider the following instance: of the $n$ agents, $n-1$ are *private* agents each having $k$ private items that have valuation 1 for the corresponding private agent and 0 for every other agent. The $n$-th agent is a *public* agent and there are $k$ public items that have a valuation of 1 for all the $n$ agents.

The optimal solution assigns the private items to the corresponding private agents and the public items to the public agent. Thus, OPT $= k$. In the online instance, the adversary chooses to present all the public items before the private items. Since all the agents look identical before the arrival of the first private item, the public agent gets no more than $k/n$ items in expectation for any algorithm (this adversarial input is comparable to our toy example discussed in Figure 1). Since none of the remaining private items can be allocated to the one public agent, their bundle value can no longer increase beyond $k/n$. The theorem follows. $\qquad\square$

**Theorem A.1.** *In the adversarial setting, for any $\varepsilon \in (0, 1)$, there is an algorithm for the online Santa Claus problem that has a competitive ratio of $(1 - \varepsilon) \cdot \frac{1}{n}$ for OPT $> C \cdot \frac{n \ln n}{\varepsilon^2}$ for a large enough constant $C$.*

*Proof.* The algorithm is to simply assign every item uniformly at random among all the $n$ agents. Note that for any fixed agent, its expected value is at least $\frac{1}{n} \cdot$ OPT. By Chernoff bounds, the probability that its total value is less than $(1 - \frac{\varepsilon}{2})\frac{1}{n} \cdot$ OPT is given by $\exp(-\frac{1}{2} \cdot \frac{\varepsilon^2}{4} \cdot \frac{1}{n} \cdot \text{OPT}) < \frac{\varepsilon}{2n}$ for OPT $> C \cdot \frac{n \ln n}{\varepsilon^2}$ for a sufficiently large constant $C$. By union bound over all the $n$ agents, the probability that *any* agent's overall value is less than $(1 - \frac{\varepsilon}{2})\frac{1}{n} \cdot$ OPT is at most $\frac{\varepsilon}{2}$. Thus, the expected competitive ratio is at least $(1 - \frac{\varepsilon}{2})(1 - \frac{\varepsilon}{2}) \cdot \frac{1}{n} > (1 - \varepsilon) \cdot \frac{1}{n}$. $\qquad\square$

### A.2  Online Algorithm Lemmas

*Proof of Lemma 2.3.* Property $(a)$ is an established property of the LogSumExp function [14, 53] but we present the proof here for completeness. We start by exponentiating the input, summing over all elements and applying the logarithm to the resultant bounds.

$$\exp \max_i \{-\varepsilon u_i\} \le \sum_{i=1}^{n} \exp -\varepsilon u_i < n \exp \max_i \{-\varepsilon u_i\}$$

$$\overset{(I)}{\Longleftrightarrow} \max_i \{-\varepsilon u_i\} \le \ln \sum_{i=1}^{n} \exp -\varepsilon u_i < \max_i \{-\varepsilon u_i\} + \ln n$$

$$\overset{(II)}{\Longleftrightarrow} -\max_i \{-\varepsilon u_i\} > -\ln \sum_{i=1}^{n} \exp -\varepsilon u_i \ge -\max_i \{-\varepsilon u_i\} - \ln n$$

$$\overset{(III)}{\Longleftrightarrow} \min_i \{\varepsilon u_i\} > -\ln \sum_{i=1}^{n} \exp -\varepsilon u_i \ge \min_i \{\varepsilon u_i\} - \ln n$$

Where (I) is the result of taking the logarithm, (II) is a negation on the inequalities and (III) is by property of the maximum. Now, since $\varepsilon > 0$, the result follows from simple algebraic manipulation.

$$\min_i \{\varepsilon u_i\} > -\ln \sum_{i=1}^{n} \exp -\varepsilon u_i \ge \min_i \{\varepsilon u_i\} - \ln n$$

$$\overset{(IV)}{\Longleftrightarrow} \varepsilon \min_i \{u_i\} > -\ln \sum_{i=1}^{n} \exp -\varepsilon u_i \ge \varepsilon \min_i \{u_i\} - \ln n$$

$$\overset{(V)}{\Longleftrightarrow} \ \min_i\{u_i\} > \frac{-1}{\varepsilon}\ln\sum_{i=1}^{n}\exp-\varepsilon u_i \geq \min_i\{u_i\} - \frac{\ln n}{\varepsilon}$$

$$\overset{(VI)}{\Longleftrightarrow} \ \min_i\{u_i\} > \phi_\varepsilon(u) \geq \min_i\{u_i\} - \frac{\ln n}{\varepsilon}$$

where (IV) follows from positive scalar multiplication within a minimum, (V) by dividing through by $\varepsilon$ and (VI) is merely the definition of our smoothing function $\phi_\varepsilon$. This verifies the desired property.

To prove $(b)$, we first calculate the partial derivative of the smoothed minimum function

$$\frac{\partial}{\partial x_i}\phi_\varepsilon(u) = \frac{e^{-\varepsilon u_i}}{\sum_{j=1}^{n}e^{-\varepsilon u_j}}$$

and now, using $u_i \geq 0$ and $v_i \in [0,1]$ we derive

$$\frac{e^{-\varepsilon}e^{-\varepsilon u_i}}{\sum_{j=1}^{n}e^{-\varepsilon u_j}} < \frac{e^{-\varepsilon(u_i+v_i)}}{\sum_{j=1}^{n}e^{-\varepsilon(u_j+v_j)}} < \frac{e^{-\varepsilon u_i}}{e^{-\varepsilon}\sum_{j=1}^{n}e^{-\varepsilon u_j}}$$

$$e^{-\varepsilon}\frac{\partial}{\partial x_i}\phi_\varepsilon(u) < \frac{\partial}{\partial x_i}\phi_\varepsilon(u+v) < e^{\varepsilon}\frac{\partial}{\partial x_i}\phi_\varepsilon(u)$$

Therefore, we have property $(b)$.

Lastly, to prove $(c)$ we first invoke the definition $\phi_\varepsilon$:

$$\frac{-1}{\varepsilon}\ln\left(\sum_{i=1}^{n}e^{-\varepsilon(x_i-y_i)}\right) \leq \frac{-1}{\varepsilon}\left(\ln\left(\sum_{i=1}^{n}e^{-\varepsilon x_i}\right) - \ln\left(\sum_{i=1}^{n}e^{-\varepsilon y_i}\right)\right).$$

This statement is equivalent to

$$\ln\left(\sum_{i=1}^{n}e^{-\varepsilon(x_i-y_i)}\right) \geq \ln\left(\sum_{i=1}^{n}e^{-\varepsilon x_i}\right) - \ln\left(\sum_{i=1}^{n}e^{-\varepsilon y_i}\right)$$

which, by exponentiating both sides, yields

$$\sum_{i=1}^{n}e^{-\varepsilon(x_i-y_i)} \geq \left(\sum_{i=1}^{n}e^{-\varepsilon x_i}\right)\cdot\left(\sum_{i=1}^{n}e^{-\varepsilon y_i}\right)^{-1} \Rightarrow \sum_{i=1}^{n}e^{-\varepsilon(x_i-y_i)}\cdot\left(\sum_{i=1}^{n}e^{-\varepsilon y_i}\right) \geq \left(\sum_{i=1}^{n}e^{-\varepsilon x_i}\right)$$

and expansion of the left-hand side verifies the claim. $\square$

*Proof of Lemma 2.4.* By direct integration and the stability property $(b)$, we see

$$\phi_\varepsilon(u+v) = \phi_\varepsilon(u) + \int_0^1 \langle\nabla\phi_\varepsilon(u+\alpha v), v\rangle d\alpha$$

$$\in \phi_\varepsilon(u) + e^{\pm\varepsilon}\langle\nabla\phi_\varepsilon(u), v\rangle$$

Therefore, we can further show

$$\langle\nabla\phi_\varepsilon(u), v\rangle \geq e^{-\varepsilon}[\phi_\varepsilon(u+v) - \phi(u)]$$
$$\geq e^{-\varepsilon}[\phi_\varepsilon(u+v') - \phi(u)]$$
$$\geq e^{-2\varepsilon}\langle\nabla\phi_\varepsilon(u), v'\rangle$$

where the first and last inequality is a direct result of Lemma 2.3, and the second is from assumption on the inputs. $\square$

*Proof of Lemma 2.5.* Let $\mu = \frac{1}{m}\sum_t y^t$ be the average of the set of vectors and to simplify notation we additionally break apart the expectation and let $\mathbb{E}\langle\mathbf{Y}^k, \mathbf{Z}\rangle = \mathbb{E}\mathbb{E}_{k-1}\langle\mathbf{Y}^k, \mathbf{Z}\rangle$ where $\mathbb{E}_{k-1}$ denote the expectation conditioned on $\mathbf{Y}^1, ..., \mathbf{Y}^{k-1}$. Note that since $\mathbf{Z}$ is a unit-vector in $\ell_1^+$, an innerproduct of this vector with $\mathbf{Y}^k$ is simply a weighted sum of the latter's elements. Thus, we have

$$\mathbb{E}_{k-1}\langle\mathbf{Y}^k, Z\rangle \geq \min_i\{\mathbb{E}\left[\mathbf{Y}^k|\mathbf{Y}^1, ..., \mathbf{Y}^{k-1}\right]\}. \tag{4}$$

Additionally, by nature of the sampling set and the procedure of sampling without replacement, we have the conditional expectation

$$\mathbb{E}\left[\mathbf{Y}^k | \mathbf{Y}^1, ..., \mathbf{Y}^{k-1}\right] = \frac{m\mu - (\mathbf{Y}^1 + ... + \mathbf{Y}^{k-1})}{m - (k-1)}$$

and further note that $m\mu - (\mathbf{Y}^1 + ... + \mathbf{Y}^{k-1})$ has the *same* distribution as $\mathbf{Y}^1 + ... + \mathbf{Y}^{m-(k-1)}$. This concretely gives us the simplifying equivalences

$$\mathbb{E}\left[\mathbf{Y}^k | \mathbf{Y}^1, ..., \mathbf{Y}^{k-1}\right] = \frac{m\mu - (\mathbf{Y}^1 + ... + \mathbf{Y}^{k-1})}{m - (k-1)} = \frac{\sum_{t=1}^{m-(k-1)} \mathbf{Y}^t}{m - (k-1)}.$$

We now return to the inequality bound of (4) and, using the above equivalences, obtain

$$\mathbb{E}\left[\min_i \left\{\sum_{t=1}^{m-(k-1)} \mathbf{Y}_i^t\right\}\right] \geq \mathbb{E}\left[\phi_\varepsilon\left(\Sigma_{t=1}^{m-(k-1)}\mathbf{Y}^t\right)\right] \qquad \text{(Lemma 2.3)}$$

$$= \mathbb{E}\left[\frac{-1}{\varepsilon}\ln\left(\sum_i \exp\left(-\varepsilon\Sigma_{t=1}^{m-(k-1)}\mathbf{Y}_i^t\right)\right)\right]$$

$$\geq \frac{-1}{\varepsilon}\ln\left(\sum_i \exp\left(-\varepsilon\mathbb{E}\left[\Sigma_{t=1}^{m-(k-1)}\mathbf{Y}_i^t\right]\right)\right) \qquad \text{(Jensen's Ineq.)}$$

$$\geq e^{-\varepsilon}\min_i\left\{\mathbb{E}[\Sigma_{t=1}^{m-(k-1)}\mathbf{Y}_i^t]\right\} - \frac{\ln n}{\varepsilon} \qquad \text{(Lemma 2.3)}$$

$$\geq e^{-\varepsilon}\min_i\left\{(m-(k-1))\mathbb{E}\mathbf{Y}_i^t\right\} - \frac{\ln n}{\varepsilon}$$

Finally combining the above results, we obtain

$$\frac{1}{m-(k-1)}\left(e^{-\varepsilon}\min_i\{(m-(k-1))\mathbb{E}\mathbf{Y}^t\} - \frac{\ln n}{\varepsilon}\right) \leq \mathbb{E}\left[\min_i\{\mathbb{E}\left[\mathbf{Y}_i^k|\mathbf{Y}^1, ..., \mathbf{Y}^{k-1}\right]\}\right]$$

$$\leq \mathbb{E}\langle\mathbf{Y}^t, \mathbf{Z}\rangle$$

Thus, after rearranging terms, this completes the lemma. $\qquad\square$