# OpenReview forum: "Online Algorithms for the Santa Claus Problem"
_NeurIPS.cc/2022/Conference — NeurIPS 2022 Accept_

### Official Review · Reviewer_TiYu · 2022-07-04

**Rating:** 6
**Confidence:** 4
**Soundness:** 2 fair
**Presentation:** 4 excellent
**Contribution:** 2 fair

**Summary:**

This paper studies the maxmin fair allocation problem in the online setting where items have to be assigned immediately upon arrival. The players have additive valuations and the goal is to maximize the minimum valuation accross all players.

The author first show that in adversarial order, not much can be done: no algorithm has competitive ratio better than 1/n where n is the number of players. This motivates the fact that one has to relax the model a little bit to hope for something interesting. The authors then consider the random arrival order. In this setting, they give a greedy-like algorithm that guarantees a competitive ratio $1-\epsilon$ under the assumption that OPT is at least $\Omega(\log(n)/\epsilon^2)$ (item have values at most $1$). Finally, they show that if OPT is not at least $\Omega(\log(n)/\epsilon)$ then it is impossible to obtain such a guarantee. Hence there result is tight up to one extra $1/\epsilon$.

**Questions:**

First I have a few questions regarding the proof of Theorem 3.1.

- Line 215: Could you elaborate on how you use the concavity in the second line? To me it seems you use the fact that $\phi$ is superadditive but shouldn't it be subadditive?

- Line 223: Should you replace $\phi_\epsilon (V^t)\geq \phi_\epsilon (\overline{V^t})$ by $\phi_\epsilon (\sum_{\tau=1}^t V^\tau)\geq \phi_\epsilon (\sum_{\tau=1}^{t-1} V^\tau+ \overline{V^t})$ ?

- Line 223-227: you say that "$\nabla^t$ is independent of $V^t$" but it seems that you need  "$\nabla^t$ is independent of $\overline{V^t}$" to use Lemma 3.5. And also either way, why is this independence property true?

- Line 223-227: More generally, could you elaborate more on the setting of the random experiment and how you apply Lemma 3.5?

Minor comments/questions:

- Maybe specify more clearly that you assume that all items have valuation at most 1. Otherwise the assumption that OPT is at least $\log(n)$ does not make much sense.

- Line 168: you say that splitting the instance in 2 halves is "essential for reducing correlations". Is this technique actually necessary? Or could it be that without splitting in half it would still work?

- References [2],[5],[11],[20],[21],[23],[40] appear only with the year but not the conference/journal it was published in.




**Strengths And Weaknesses:**

The strengths of the paper are (in my opinion) the following:

- the problem is interesting and this is an important problem in the field of approximation algorithms. It is nice to see this problem studied in the online model.

- the results are almost tight.

- Overall the paper is well-written and easy to read.

The weaknesses are (in my opinion) the following:

- the techniques do not seem really surprising.

- some parts of the proof of the main result are unclear to me (see questions to clarify). I am a little bit worried about this part but I am willing to change my score if the authors clarify.

- Although the results are tight, I am not sure exactly of how significant they are. This assumption that OPT is at least log(n) makes the results a little bit less interesting in my opinion (even though the authors show this is necessary).

---

> ### Author Response · Authors · 2022-08-02
> **Response**
>
> We thank the reviewer for their careful reading of our analysis and noting portions that could be expanded upon for clarity. We here address the questions raised and hope to alleviate any issues.
>
> The restart at $n/2$ of the algorithm is essential to reduce correlations between the arriving inputs: the standard greedy algorithm's allocation selection will depend on at most $n-1$ prior decisions, rather than $n/2 - 1$. Notably, we suspect the standard greedy algorithm in general yields a lower competitive ratio in the random order model than with the restart (as is the case for the dual makespan problem) but have here omitted a theoretical treatment of this fact, instead citing the work of Gollapudi [CIKM'14] and Gupta [ESA'14].
>
> With respect to the comments on lines 215 and 223, both come from the following utilization of concavity:
>
> Claim: For $\varepsilon > 0$ and $x,y \in [0,1]^n$ with $x_i \geq y_i \geq 0$, we have $$\phi_{\varepsilon}(x-y) \leq \phi_{\varepsilon}(x) - \phi_{\varepsilon}(y)$$
>
> Proof:
> By definition of $\phi_{\varepsilon}$, the inequality becomes $$\frac{-1}{\varepsilon} \ln \left( \sum_{i=1}^n e^{-\varepsilon(x_i - y_i)} \right) \leq \frac{-1}{\varepsilon} \left( \ln \left( \sum_{i=1}^n e^{-\varepsilon x_i} \right) - \ln \left( \sum_{i=1}^n e^{-\varepsilon y_i} \right) \right).$$
>
> This statement is equivalent to $$\ln \left( \sum_{i=1}^n e^{-\varepsilon(x_i - y_i)} \right) \geq \ln \left( \sum_{i=1}^n e^{-\varepsilon x_i} \right) - \ln \left( \sum_{i=1}^n e^{-\varepsilon y_i} \right)$$ which by exponentiating both sides yields $$\sum_{i=1}^n e^{-\varepsilon(x_i - y_i)} \geq \left(\sum_{i=1}^n e^{-\varepsilon x_i} \right) \cdot \left( \sum_{i=1}^n e^{-\varepsilon y_i} \right)^{-1} \Rightarrow \sum_{i=1}^n e^{-\varepsilon(x_i - y_i)} \cdot \left( \sum_{i=1}^n e^{-\varepsilon y_i} \right) \geq \left(\sum_{i=1}^n e^{-\varepsilon x_i} \right)$$ and expansion of the left-hand side verifies the claim. $\square$
>
> As a result of this proposition, we have in line 215 that
> \begin{align*}
>     \phi_\varepsilon\left(\sum_{\tau=1}^{m/2}\mathbf{V}^{\tau}\right) + \phi_\varepsilon\left(\sum_{\tau=m/2 + 1}^{m}\mathbf{V}^{\tau}\right) &\leq \phi_\varepsilon\left(\sum_{\tau=1}^{m/2}\mathbf{V}^{\tau}\right) + \phi_\varepsilon\left(\sum_{\tau=1}^{m}\mathbf{V}^{\tau}\right) - \phi_\varepsilon\left(\sum_{\tau=1}^{m/2}\mathbf{V}^{\tau}\right) = \phi_\varepsilon\left(\sum_{\tau=1}^{m}\mathbf{V}^{\tau}\right)
> \end{align*}
> This step was omitted from the submission due to space constraints but will be added for clarity to the camera-ready version.
>
> For the comments pertaining to lines 223, we note that $\nabla^t$ is defined to be $\nabla \phi_\varepsilon(\mathbf{V}^{t-1})$ which is independent of $\mathbf{V}^t$ (the same applies for $\overline{\nabla}^t$ and $\overline{\mathbf{V}}^t$).
> We apologize if the notation was not made clear and will work to improve the readability in the final version of the paper.
>
> Lastly, the random experiment in Lemma 3.5 follows the technique of Gupta [ESA'14] and is utilized to bound the regret in optimal decision making according to the smoothed minimum as opposed to the true minimum in the random order input stream. Unlike the simpler iid setting, the random-order model introduces the difficulty of having a \emph{correlated} input stream, and the probabilistic lemma accounts for when $\mathbf{Z}$ is both highly or not correlated to the $\mathbf{Y}$. Using this and the gap between the optimal and algorithmic solution yields the final competitive ratio.

---

> > ### Comment · Reviewer_TiYu · 2022-08-04
> > **answer**
> >
> > I thank the authors for answering my technical concerns. I now feel more comfortable to raise my score from 3 to 6. The paper is nice but the fact that techniques do not seem very surprising/novel still are a weakness in my opinion.

---

### Official Review · Reviewer_QYda · 2022-07-07

**Rating:** 6
**Confidence:** 3
**Soundness:** 2 fair
**Presentation:** 3 good
**Contribution:** 3 good

**Summary:**

The paper considers the online version of a scheduling problem known as the Santa Claus Problem. Similarly as in unrelated machine scheduling, here there are jobs whose running times are machine-dependent. However, rather than minimizing makespan, the goal is to maximize the least loaded machine. The new feature in this paper is that an online model, in which jobs arrive over time, is considered. It is straightforward to observe that if the arrival order is adversarial then, not much can be done, so the paper is mostly about the random order model. The main result of the paper is an algorithm that generates a solution within a factor 1-eps of the offline optimum minus an additive term of log(n)/eps. This is when all values (or processing times) are between 0 and 1 and n is the number of machines. The result implies that if OPT is sufficiently large (say log(n)/eps^2) then the algorithm is truly 1-eps competitive. This result is complemented by Thm 1.5 stating that if OPT is smaller than log(n)/eps then a 1-eps approximation is not possible.



**Questions:**

This is a theory paper and as such, I would like the authors to clearly state what are the key technical innovations. My understanding is that the key is to use LOGSUMEXP as a smoothed minimum. But, are you the first to do this?

The intuitive explanation of the restarting in your algorithm is not clear to me. Can you elaborate?

You mention at the end that you would expect your results to somehow extend to the MinMax objective (which is probably more natural). Do you have further thoughts? while I agree that there are similarities in the problems they are still very different.

**Ethics Review Area:**

["I don’t know"]

**Limitations:**

no comments

**Strengths And Weaknesses:**

The algorithm is clever. It first finds an online fractional assignment by greedily maximizing the min load... but here instead the authors use the log of the sum of exp as smoothing. Then they note that simple randomized rounding can be used to obtain an integer solution online. While I am not completely sure, I think the algorithmic idea is not novel. Still, making it work precisely requires nontrivial work.

It is nice to have a nearly tight lower bound and the construction here is nice and simple.

---

> ### Author Response · Authors · 2022-08-02
> **Response**
>
> We thank the reviewer for their compliments on the submission and its contribution to the field. This is indeed the first result on the online Santa Claus problem under a random-order input assumption with the algorithm being adapted from prior work on the dual makespan problem.
>
> The restart at $n/2$ of the algorithm is essential to reduce correlations between the arriving inputs: the standard greedy algorithm's allocation selection will depend on at most $n-1$ prior decisions, rather than $n/2 - 1$. Notably, we suspect the standard greedy algorithm in general yields a lower competitive ratio in the random order model than with the restart (as is the case for the dual makespan problem) but have here omitted a theoretical treatment of this fact, instead citing the work of Gollapudi [CIKM'14] and Gupta [ESA'14].
>
> Lastly, our mention of extensions to the dual problem were in reference to our lower-bound instance constructions. We imagine such challenging random-order inputs can be reformulated to further analyze the tightest upper bounds on the makespan problem.

---

### Official Review · Reviewer_42k9 · 2022-07-11

**Rating:** 6
**Confidence:** 4
**Soundness:** 4 excellent
**Presentation:** 3 good
**Contribution:** 3 good

**Summary:**

The paper considers the online version a scheduling problem known as the Santa Claus problem. one is given m jobs that are to be distributed (integrally) among n processors. Each job has a given non-negative value for each machine that is obtained if the job is assigned to that machine. The goal is to assign the jobs so as to maximise the minimum total value of the jobs assigned to any machine. Since the paper considers the online variant of the problem, the jobs arrive one by one and the algorithm must assign each job irrevocably without prior knowledge of jobs that will arrive in the future.

It can be easily seen that if the arrival order of the jobs is adversarial, then any algorithm performs arbitrarily bad in the worst case. The paper however shows that under the assumption that there is a specific lower bound on the value of the optimal assignment, then there exists an algorithm with a competitive ratio arbitrarily close to 1, and this is "essentially" tight. The algorithm is based on first producing a fractional assignment in two stages the first stage is "greedy" and also aims to "observe" the input instance before trying to maximise the objective further in the second stage. A rounding scheme is provided to produce an integral assignment.

**Questions:**

- Why is this particular problem of interest to the NeuRIPS community?

**Limitations:**

No ethical considerations.

**Strengths And Weaknesses:**

The paper is very well written, the proofs are non-trivial and the obtained results as well as techniques employed are definitely of interest. To summarise, this is overall  a very nice paper.

Nevertheless, I hesitate to give a stronger score because it is not clear to me (and neither argued in the paper as far as I can tell) why these results within scope for NeuRIPS and of interest to this particular community.

Minor comments:
- I would prefer "uniformly random" instead of just "random". Same on definition 1.2, how are the vectors "randomly permuted"?
- Lines 38-39. You do mention before that the Santa Claus problem was studied before but this sentence seems to suggest that perhaps you introduce it.
- I understand that the assumption on the lower bound regarding the value that OPT assigns to any machine is necessary, but it would be nice to also provide some intuition on where and how exactly it is used.

---

> ### Author Response · Authors · 2022-08-02
> **Question Response**
>
> We thank the reviewer for their kind words regarding the writing of our paper and presentation of the results. We defer them to the above general comments for emphasis on our work's place at NeurIPS.

---

> > ### Comment · Reviewer_42k9 · 2022-08-08
> > **Ack**
> >
> > Thanks for the response. I would like to point out that at least part of the example papers that you list have a connection to ML via their use of machine learned predictions, which is not the case with the current paper. Still, as I wrote this is a good paper and I am not opposed to accepting it.

---

### Official Review · Reviewer_pZpD · 2022-07-12

**Rating:** 6
**Confidence:** 4
**Soundness:** 4 excellent
**Presentation:** 3 good
**Contribution:** 2 fair

**Summary:**

The paper considers an online version of the well studied Santa Claus problem - a classical problem in fair division. There are n agents and m items that arrive online. When an item arrives, its value for each agent is revealed and the algorithm needs to assign the item to one of the agents. The goal of the algorithm is to find an allocation to maximize the minimum total value obtained by any agent.
The paper proposes a (1-epsilon)-competitive algorithm for this problem under two assumptions, (i) the items arrive in a uniformly random order, and (ii) the instance is sufficiently large, i.e., the optimal solution has a value of Omega(log n/epsilon^2).


**Questions:**

None

**Strengths And Weaknesses:**

The Santa Claus problem is a well-studied problem in fair division and the online generalization studied here is very natural and important. The paper is well-written and is fairly easy to read and follow.

The algorithm presented is an adaptation of Molinaro’s algorithm for online scheduling for load balancing and is a natural greedy algorithm that assigns the online item so as to maximize a “smoothed” minimum of the agent valuations. The key insight (adapted from prior work) is to effectively restart the algorithm after half the items have been seen so as to reduce the correlations between arriving items.

The paper also provides a lower bound to demonstrate that the additive factor in the analysis (or the lower bound on OPT) is necessary to obtain a (1-\epsilon)- competitive algorithm even in the random arrival model.

I am not fully convinced of the relevance of the subject matter to Neurips however. I feel that the paper would be more at home at a theoretical venue.

---

> ### Author Response · Authors · 2022-08-02
> **Question Response**
>
> We thank the reviewer for their warm comments with respect to our paper's presentation and overall results. We defer them to the above general comments for emphasis on our work's place at NeurIPS.

---

### Author Response · Authors · 2022-08-02
**Overall Comments**

We thank the reviewers for their helpful comments. In particular, we appreciate the noting of typos, citation and grammatical errors -- all of which have been rectified for the camera-ready version of our submission. We here address the comments concerning the \textit{soundness} and \textit{contribution} of our paper.

With respect to concerns over the relevance of our paper to NeurIPS, we note that the theoretical nature of our work and its area of focus has been highly prevalent at previous iterations of the conference. Work in online algorithms with upper and lower bound constructions dates back as far as "An Online Algorithm for Maximizing Submodular Functions'' [NIPS'08] with more recent papers including "Group-Fair Online Allocation in Continuous Time'' [NIPS'20], "Optimal Robustness-Consistency Trade-offs for Learning-Augmented Online Algorithms'' [NIPS'20], and "Online Knapsack with Frequency Predictions'' [NIPS'21] to name just a few. Moreover, as a direct precursor of our work, the paper ``Fair Scheduling for Time-dependent Resources'' appeared in NeurIPS '21 which examined allocation procedures under a maxmin objective in the \textit{offline} problem setting. As such, the nearly tight result we have presented here will be of substantial interest to attendees and future readers of the conference proceedings in the field of algorithmic game theory and beyond.

We further highlight the implications of our algorithmic results for ensuring \textit{fairness} in online systems. Our presented results ensure in this online setting (as is the case in ad auctions) that all individuals receive an equitable share of the available resources. This notion of fair algorithms is highly prevalent at NeurIPS (two recent examples being "Fair Clustering Under a Bounded Cost'' [NIPS'21] and "Fair Hierarchical Clustering'' [NIPS'20]), and our work provides an important result in the less explored online problem context.

---

### Meta-Review · Area_Chair_Xkqj · 2022-08-23

**Recommendation:** Accept
**Confidence:** Less certain

**Metareview:**

The Santa Claus problem is a well-known problem in fair division and the online setting considered in this paper very natural and interesting paper. The reviewers found the paper well-written and interesting. We encourage the authors to incorporate the reviewers comments into their final version. Further, given this problem is less studied in NeurIPS community, please think of motivating the problem for broader NeurIPS audience.

**Award:**

No

---

### Decision · Program_Chairs · 2022-09-14

Accept